# Iridium oxide nanoribbons with metastable monoclinic phase for highly efficient electrocatalytic oxygen evolution

Fan Liao [1,4], Kui Yin[1,2,4], Yujin Ji[1,4], Wenxiang Zhu[1], Zhenglong Fan[1], Youyong Li [1], Jun Zhong[1], Mingwang Shao [1], Zhenhui Kang [1,3] & Qi Shao [2] ✉

Metastable metal oxides with ribbon morphologies have promising applications for energy conversion catalysis, however they are largely restricted by their limited synthesis methods. In this study, a monoclinic phase iridium oxide nanoribbon with a space group of C2/m is successfully obtained, which is distinct from rutile iridium oxide with a stable tetragonal phase (P42/mnm). A molten-alkali mechanochemical method provides a unique strategy for achieving this layered nanoribbon structure via a conversion from a monoclinic phase $K_{0.25}IrO_2$ (I2/m (12)) precursor. The formation mechanism of $IrO_2$ nanoribbon is clearly revealed, with its further conversion to $IrO_2$ nanosheet with a trigonal phase. When applied as an electrocatalyst for the oxygen evolution reaction in acidic condition, the intrinsic catalytic activity of $IrO_2$ nanoribbon is higher than that of tetragonal phase $IrO_2$ due to the low $d$ band centre of Ir in this special monoclinic phase structure, as confirmed by density functional theory calculations.

Electrolyzing water into clean and renewable hydrogen in acidic electrolytes is an effective technique to relieve energy and environmental protection concerns[1–4]. An efficient electrocatalyst for an anodic oxygen evolution reaction (OER) significantly improves the energy conversion efficiency because most of the energy consumption of electrolytic water splitting occurs at the anode[5–7]. Iridium oxide ($IrO_2$) is considered to be a state-of-the-art catalyst for OER that can withstand harsh acidic condition[8–11]. However, the previously reported $IrO_2$ catalysts lack high intrinsic activity against the slow kinetics of OER.

Constructing a metastable nanostructure is a promising method to pursue superior catalytic properties[12–17]. Naturally occurring thermodynamically stable $IrO_2$ is commonly found in a rutile phase with pristine regular [Ir-$O_6$] units linked by shared edge and corner modes. Earlier works have demonstrated that the intrinsic activity of $IrO_2$ is closely associated with [Ir-$O_6$] unit linkage construction[18–21] and lattice distortion in [Ir-$O_6$] octahedrons[22–24]. Therefore, designing a

metastable nanostructure with different unit linkages may provide a completely distinct active surface for electrocatalysis and a deep understanding of the structure-activity relationship[25–29]. Low-dimensional materials with increased surface energies provide substrates for the development of advanced catalysts due to their inherent anisotropic properties, quantum confinement characteristics and edge effects[30,31]. However, $IrO_2$ with an unconventional crystal phase and nanoribbon morphology has not yet been reported.

Motivated by all these possibilities, in this work, a metal oxide comprising monoclinic phase layered $IrO_2$ nanoribbons ($IrO_2$NRs), with the space group C2/m (12), is prepared. The Ir-O coordination polyhedrons in the $IrO_2$NRs maintain an octahedral configuration, while the connections of the octahedral subunits are in an edge-sharing mode. The space group of the $IrO_2$NR is C2/m (12), which is totally different from that of Rutile $IrO_2$ (P4$_2$/mnm (136)). A molten-alkali mechanochemical method promotes metastable phase

[1]Institute of Functional Nano & Soft Materials (FUNSOM), Jiangsu Key Laboratory for Carbon-Based Functional Materials & Devices, Soochow University, 199 Ren'ai Road, Suzhou 215123 Jiangsu, China. [2]College of Chemistry, Chemical Engineering and Materials Science, Soochow University, Suzhou, Jiangsu 215123, China. [3]Macao Institute of Materials Science and Engineering, Macau University of Science and Technology, Taipa, 999078 Macau, SAR, China. [4]These authors contributed equally: Fan Liao, Kui Yin and Yujin Ji. ✉e-mail: qshao@suda.edu.cn

formation from a precursor of monoclinic $K_{0.25}IrO_2$ (I2/m (12)). The formation mechanism of the $IrO_2NR$ and the further conversion of $IrO_2$ nanosheets ($IrO_2NSs$) are clearly determined according to an experimental observation. Due to the specific phase construction and nanoribbon morphology, the $IrO_2NR$ exhibits superior OER activity and stability in acidic electrolytes. Theoretical calculations have been carried out to explain the high intrinsic activity of $IrO_2NR$.

## Results

### Morphology and structure of IrO₂NRs

$IrO_2NRs$ are fabricated via a molten-alkali mechanochemical method in a homemade mechano-thermal reactor, which provides unique synthetic conditions, such as high temperature, a strong alkaline media and a continuous grinding force. The fabrication process is schematically shown in Supplementary Fig. 1. The raw materials ($IrCl_3$ and KOH) are first stirred continuously at 150 °C in a Teflon mortar. Then, a dark blue slurry forms and is transferred to the homemade mechano-thermal reactor, where it is heated at 700 °C for 2 h to generate $IrO_2NRs$.

The morphologies of the $IrO_2NRs$ on a large scale are first observed by scanning electron microscopy (SEM) as shown in Fig. 1a. All the samples exhibit a uniform nanoribbon structure. The dense and flexible $IrO_2NRs$ are entangled. The transmission electron microscopy

(TEM) image shown in Supplementary Fig. 2 displays a nanoribbon morphology at a high magnification. A typical TEM image of a single $IrO_2NR$ is shown in Fig. 1b. From the enlarged figure of the tail end, the width of this nanoribbon is 7.9 nm. The average width is 10.0 nm, based on 500 individual samples (Supplementary Fig. 3). Pictures of several distinctive $IrO_2NRs$ are displayed in Supplementary Fig. 4 to show their widths and lengths. A typical long $IrO_2NR$ is shown in Supplementary Fig. 4a, with a length of ~22.53 μm. A thin $IrO_2NR$ with a width of 4.9 nm is shown in Supplementary Fig. 4b. The lattice tension/compression can be observed at the edges of the $IrO_2NRs$ according to the HRTEM images (Supplementary Fig. 5). The average lattice strain is calculated by the Williamson-Hall equation based on the XRD data (Supplementary Fig. 6), which is about 0.388%. Although the lattice strain exists in $IrO_2NR$, the effect on activity may be limited due to its low value when we compared with the stain in previous reference about $IrO_2$ for OER catalysis[32]. The thicknesses of the $IrO_2NRs$ range from ~6.0 to 12.0 nm, as confirmed by atomic force microscopy (AFM) images (Supplementary Fig. 7). The resistivity of the $IrO_2NR$ is estimated to be $1.2 \times 10^{-6}$ Ω m, indicating its high conductivity behaviour (Supplementary Fig. 8).

Notably, an X-ray powder diffractometer (XRD) pattern of $IrO_2NR$ (Fig. 1c) exhibits four peaks with proportional $d$ values, indicating that $IrO_2NR$ has a layered structure with a $c$-axis spacing of 6.95 ± 0.03 Å.

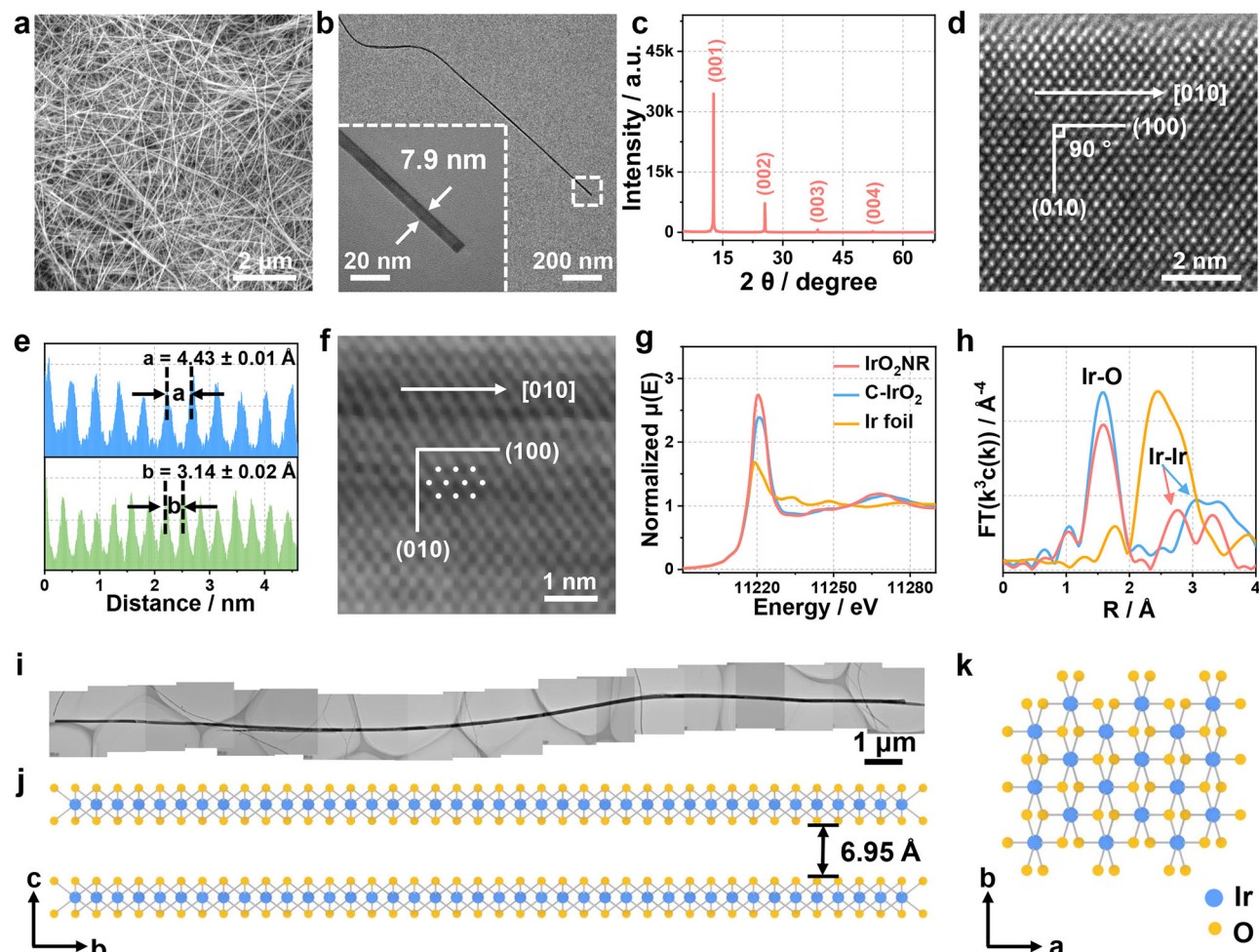

**Fig. 1 | Morphology and structure of the IrO₂NR. a** SEM image showing the uniform distribution of the $IrO_2NR$. **b** TEM image of a single $IrO_2NR$. The insert is the enlargement of the tail end of the ribbon with a width of 7.9 mm. **c** XRD pattern. **d** HRTEM image. **e** Line scan of the HRTEM image indicated by the (100) and (010) planes. **f** HAADF−STEM image for the $IrO_2NR$. **g** XANES spectra and **h** FT−EXAFS spectra for the $IrO_2NR$, $C\text{-}IrO_2$ and Ir foil. **i** A typical long $IrO_2NR$ with a length of 22.53 μm. **j, k** Pictorial illustration of the crystal structure for the $IrO_2NR$ from the (**j**) $a$ direction and (**k**) $c$ direction.

The EDS spectrum shown in Supplementary Fig. 9a indicates that the $IrO_2$NRs are composed of only Ir and O, with atomic ratios close to 1: 2. A high-angle annular dark-field scanning transmission electron microscopy (HAADF–STEM) image and a corresponding elemental mapping (Supplementary Fig. 9b–d) reveal the homogeneous distributions of Ir and O in the $IrO_2$NRs.

The high-resolution transmission electron microscopy (HRTEM) image shown in Fig. 1d displays the well-resolved atomic structure of the $IrO_2$NR. The lattice fringe spacing and the crystal plane angle are measured accordingly. Based on the selection principle of the Bravais lattices, two directions, defined as $a$ and $b$ with crystal plane angles of 90°, are selected to represent the unit cell. The parameters of the unit cell are as follows: $a = 4.43 \pm 0.01$ Å, $b = 3.14 \pm 0.02$ Å, and $\gamma = 90°$ (Fig. 1d, e). The aberration-corrected HAADF–STEM image in Fig. 1f shows the atomic arrangement of Ir, which is in accordance with that in the HRTEM image. The cell parameters of the $IrO_2$NRs are obviously different from those of Rutile $IrO_2$ (Supplementary Table 1)[33]. The selective-area electron diffraction (SAED) pattern in Supplementary Fig. 10 indicates the crystalline nature of the $IrO_2$NRs. The $IrO_2$NRs show layered structures and tend to lay flat on the copper mesh, making it difficult to find a plane that is perpendicular to the [001] direction. Then, the $IrO_2$NRs are embedded in epoxy resin and sliced into pieces, which are picked up with copper mesh. From the HRTEM image of a cross section of $IrO_2$NR (Supplementary Fig. 11), a distance of 0.69 nm that belongs to (001) plane is observed, which is in accordance with the XRD results. Additionally, due to the preferred orientation of the $IrO_2$NR, no peaks other than the layered structure peaks are observed in the XRD pattern.

The structure and oxidation states of the $IrO_2$NR are analysed by synchrotron X-ray absorption spectroscopy (XAS) using commercial $IrO_2$ (C-$IrO_2$, rutile phase) and Ir foil as the reference. An Ir-$L_3$ edge X-ray absorption near-edge spectroscopy (XANES) image of the $IrO_2$NR is shown in Fig. 1g. The absorption edge and spectral shape of the $IrO_2$NR are similar to those of C-$IrO_2$, indicating an oxidation state of +4 for Ir in the $IrO_2$NR[34]. Fourier transformed extended X-ray absorption fine structure spectra (FT–EXAFS) in Fig. 1h show the scattering profile as a function of the radial distance from the central absorbing Ir atom. The spectra here are plotted without phase correction. For the $IrO_2$NR, the main peak is located at -1.6 Å, corresponding to the Ir-O distance in the [Ir-$O_6$] octahedron, which is similar to the first coordination shell of Rutile $IrO_2$[35]. Additionally, the second Ir-Ir shell is clearly observable in the $IrO_2$NR, which is located at -2.76 Å; this location is lower than that in C-$IrO_2$, indicating that the interconnection of the [Ir-$O_6$] octahedron follows another model in the $IrO_2$NR[36]. According to the HRTEM image and the XAS data, the structure of the $IrO_2$NR (Fig. 1i) from the $a$ and $c$ directions is determined and shown in Fig. 1j, k. There is a two-fold rotation axis through the $b$ direction with a vertical reflection. The symmetry space group of the $IrO_2$NR is determined to be monoclinic C2/m (12), and the corresponding crystallographic information is shown in Supplementary Table 1. All the above results conclude that a layered $IrO_2$ nanostructure with monoclinic C2/m (12) phase is successfully obtained, the determination of which is schematically shown in Supplementary Fig. 12. A simulated XRD pattern based on the structure of the $IrO_2$NR is shown in Supplementary Fig. 13. As seen from the figure, peaks at -12 and 25° are observed. The disappearance of other peaks is due to the preferred orientation of the $IrO_2$NR. The exposed crystal faces obtained from the [001] direction in the SAED patterns (Supplementary Fig. 10) are in accordance with the stimulated XRD pattern of the $IrO_2$NR structure, confirming its novel structure.

## Formation mechanism of $IrO_2$NRs
Considering the unique phase and dimensionality property of $IrO_2$NRs, their formation processes are investigated in detail. The initial slurry obtained by heating the mixture of ethanol solution of $IrCl_3$ and KOH at 150 °C is taken out directly and characterized by XRD and TEM as shown in Supplementary Fig. 14. The XRD pattern of the slurry sample before washing is shown in Supplementary Fig. 14a. Although the initial slurry is very complex due to the strong alkaline medium and hydrate formation, the main peaks can be indexed as $KIrO_3$, KOH, $KO_{1.58}H_{0.42}$ and KOH·$H_2O$. The slurry sample shows a droplet-like morphology as seen in the TEM image in Supplementary Fig. 14b. The slurry at this stage is easily dissolved in water during the washing process. After washing, there is no solid product. The XRD patterns and morphology images of the products obtained at reaction temperatures from 300 to 600 °C are shown in Supplementary Fig. 15. When the reaction temperature ranges from 300 to 400 °C, the samples are amorphous and show no unique shapes (Supplementary Fig. 15a–f). Upon increasing the temperature to 500 °C (Supplementary Fig. 15g–i), the bulk nanoparticles become fluffy. The XRD pattern of the sample obtained at 500 °C shows monoclinic structure $K_{0.25}IrO_2$ (PDF No. 85-2185)[37]. The chemical composition of $K_{0.25}IrO_2$ obtained at a reaction temperature of 500 °C is further investigated as shown in Supplementary Fig. 16. The HRTEM image (Supplementary Fig. 16a) shows that the crystal plane of (211) of $K_{0.25}IrO_2$. The SAED pattern shows that $K_{0.25}IrO_2$ is polycrystalline (Supplementary Fig. 16b). The bright ring corresponds to the (211) plane (JCPDS No. 85-2185). The corresponding elemental mapping results indicate the existence of K, Ir and O elements (Supplementary Fig. 16c–f). When the temperature reaches 600 °C (Supplementary Fig. 15j–l), small nanoribbon structures can be found in the TEM images. The characteristic XRD peaks of $K_{0.25}IrO_2$ begin to fade, and the layered peaks gradually form and become apparent. From these data, we speculate that the $IrO_2$NR originates from $K_{0.25}IrO_2$ and transforms into $IrO_2$NR with increasing temperature.

More comparison experiments are performed to further indicate the important roles of alkaline conditions. The $K_{0.25}IrO_2$ sample (obtained at the reaction temperature of 500 °C) is washed with water to remove the KOH and then calcined at 700 °C for 2 h. The XRD pattern shows that it transforms into Rutile $IrO_2$ (Supplementary Fig. 17a), the morphology of which is a nonuniformly shaped nanoparticle, as shown in the TEM image in Supplementary Fig. 17b. Then, KOH is added to $K_{0.25}IrO_2$, and the fabrication process of the $IrO_2$NR calcined at 700 °C for 2 h is repeated. The XRD pattern of the obtained sample shows layered peaks (Supplementary Fig. 17c). The TEM image indicates that a nanoribbon morphology can form during this process (Supplementary Fig. 17d), which confirms that KOH is very important for the formation of $IrO_2$NR.

Next, the chemical states of Ir and O in the $IrO_2$NR and $K_{0.25}IrO_2$ are investigated by X-ray photoelectron spectroscopy (XPS). XPS measurements are conducted for C-$IrO_2$ and C-Ir/C for comparison. As shown in Supplementary Fig. 18a, K is present in $K_{0.25}IrO_2$. However, only Ir and O are detected in $IrO_2$NR and C-$IrO_2$, indicating the purity of our samples. Then, the Ir 4$f$ fine spectra of the four samples are deconvoluted (Supplementary Fig. 18b). For C-$IrO_2$, Ir 4$f_{7/2}$ at 62.2 eV and Ir 4$f_{5/2}$ at 65.2 eV correspond to the reported Rutile $IrO_2$, with two accompanying satellite peaks[38,39]. The oxidation states of Ir in the $IrO_2$NRs are close to C-$IrO_2$, which is 0.3 eV higher than that of $K_{0.25}IrO_2$. No metallic Ir is detected in the $IrO_2$ samples, as determined by the binding energy of Ir in C-Ir/C[40]. The O 1$s$ spectrum of $IrO_2$NR can be deconvoluted into three peaks, including lattice oxygen ($O_{Ir-O}$), unsaturated oxygen ($O_V$) and adsorbed oxygen ($O_{H_2O}$) (Supplementary Fig. 18c).

The XANES and EXAFS spectra of the $IrO_2$NRs are compared to those of $K_{0.25}IrO_2$. The oxidation state of Ir in $K_{0.25}IrO_2$ is slightly lower than that in the $IrO_2$NR (Supplementary Fig. 19a), as seen from the first derivative of the Ir-$L_3$ edge curves between the $IrO_2$NR and $K_{0.25}IrO_2$ samples (Supplementary Fig. 19b). The second Ir-Ir shell of $K_{0.25}IrO_2$ in the FT–EXAFS spectrum (Supplementary Fig. 19c) occurs due to the edge shared [Ir-$O_6$] octahedron, the location of which is close to that of

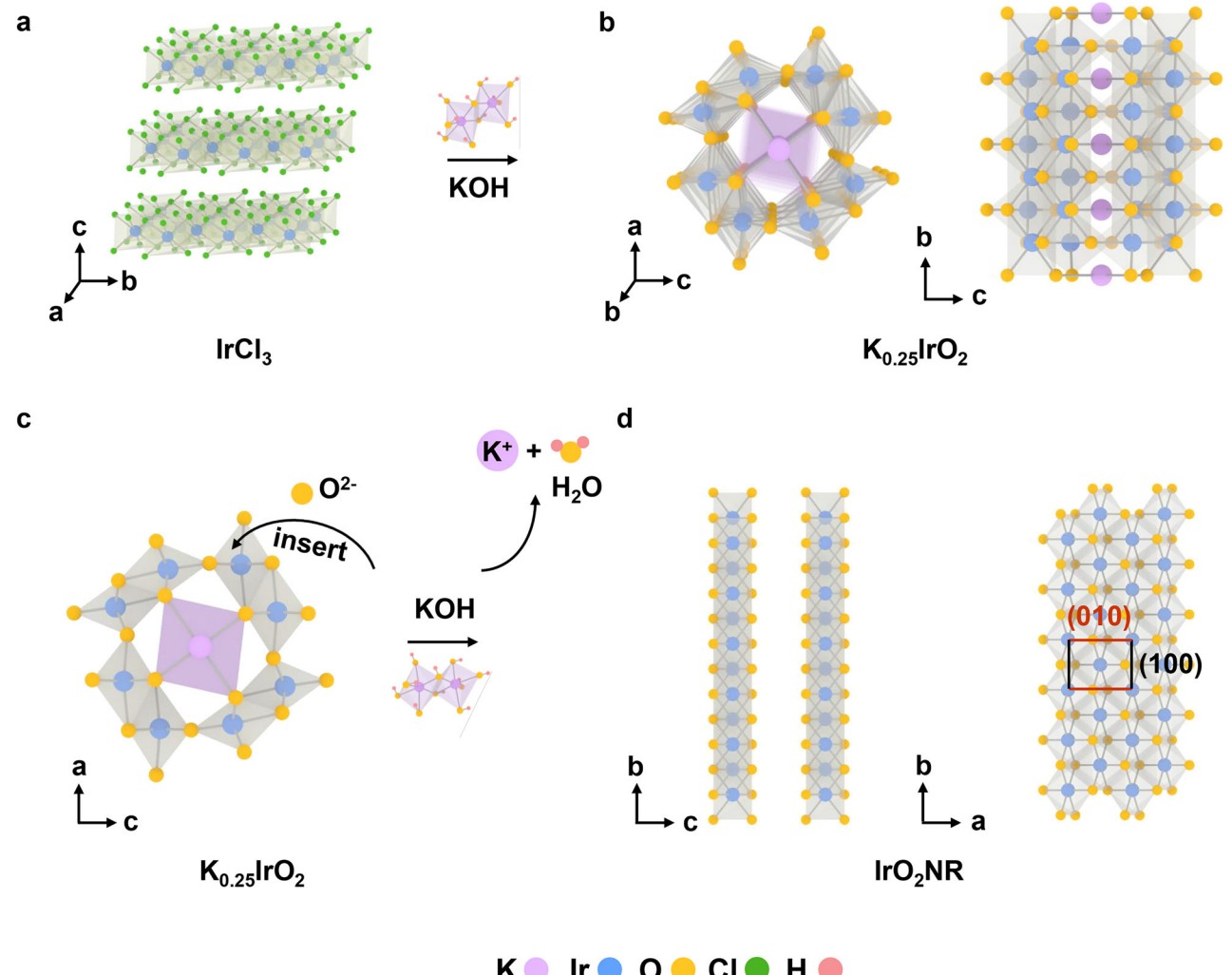

**Fig. 2 | Structural evolution of the IrO₂NR. a, b** From the raw materials of $IrCl_3$ and KOH to $K_{0.25}IrO_2$. **c, d** From $K_{0.25}IrO_2$ to the IrO₂NR. Purple, blue, yellow, green and pink balls represent K, Ir, O, Cl and H elements, respectively.

the IrO₂NR (Supplementary Fig. 19c,d)[41]. Combined with the $K_{0.25}IrO_2$ detected during the fabrication process, we conclude that the structure of the IrO₂NR evolves from that of $K_{0.25}IrO_2$ by inheriting the edge sharing [Ir-O₆] octahedron linkage mode.

Another interesting phenomenon is that when we further increase the reaction temperature to 800 °C for 2 h, most of the IrO₂NR transforms into a nanosheet morphology, which is defined as IrO₂NS (Supplementary Fig. 20). The XRD pattern of the IrO₂NS (Supplementary Fig. 20a) is similar to that of the IrO₂NR; however, that the peak intensity of IrO₂NS is larger than that of IrO₂NR, indicating that the crystallinity of IrO₂NS is higher than that of IrO₂NR, which may be due to the high reaction temperature. The SEM and TEM images in Supplementary Fig. 20b–d clearly show that most samples exhibit nanosheet morphologies with several nanoribbons. The HRTEM and SAED images (Supplementary Fig. 20e, f) show that different crystal structures between the IrO₂NS and IrO₂NR. The IrO₂NS can be assigned to trigonal phase with a space group of P-3m1 (164)[42]. The structural parameters of the IrO₂NSs are listed in Supplementary Table 1.

From the above results, the structural transformation process of this unique IrO₂NR is proposed in Fig. 2. First, $IrCl_3$ reacts with KOH by the molten-alkali mechanochemical method to form $K_{0.25}IrO_2$ (Fig. 2a, b). Then, the $O^{2-}$ ions, originating from KOH at high temperature, attack the corner-connected octahedron of $K_{0.25}IrO_2$ (Fig. 2c) to form the IrO₂NR structure (Fig. 2d). During fabrication at different

temperatures, the IrO₂NR maintains the same [Ir-O₆] edge linkage mode as $K_{0.25}IrO_2$[43]. The growth direction of the IrO₂NR is defined according to the crystal-axis orientation as $K_{0.25}IrO_2$, which means that the IrO₂NR is oriented in the *b* direction. Therefore, the (010) and (100) crystal planes are identified as marked in Fig. 2d. After the oxygen bonds break in $K_{0.25}IrO_2$, the [Ir-O₆] subunits connect through an edge-edge mode (Supplementary Fig. 21a); the number of columns of [Ir-O₆] subunits that perpendicular to the *b* direction must be even, as demonstrated by Supplementary Fig. 21b, c. The IrO₂NR retains the nanoribbon morphology in the final product because of the addition of ethanol to the raw materials. Ethanol adsorbs on the products during the fabrication process and transfers to amorphous carbon at high temperature, which slows the speed of the ribbon-to-ribbon connection. When the reaction temperature further increases to 800 °C, the edges of the IrO₂NR connect and form a nanosheet morphology with a slightly changed lattice constant, as shown in Supplementary Fig. 20g.

## Electrocatalytic activity of the IrO₂NR for OER

The OER activity of the IrO₂NR is evaluated in an O₂-saturated 0.5 M $H_2SO_4$ electrolyte using a standard three-electrode system. The $Hg/HgCl_2$ reference electrode (saturated calomel electrode; SCE) is calibrated before the OER test (Supplementary Fig. 22)[44]. C-IrO₂ and C-Ir/C are tested as benchmarks. The XRD patterns of C-IrO₂ and C-Ir/C are shown in Supplementary Fig. 23, which indicate that C-IrO₂ has a

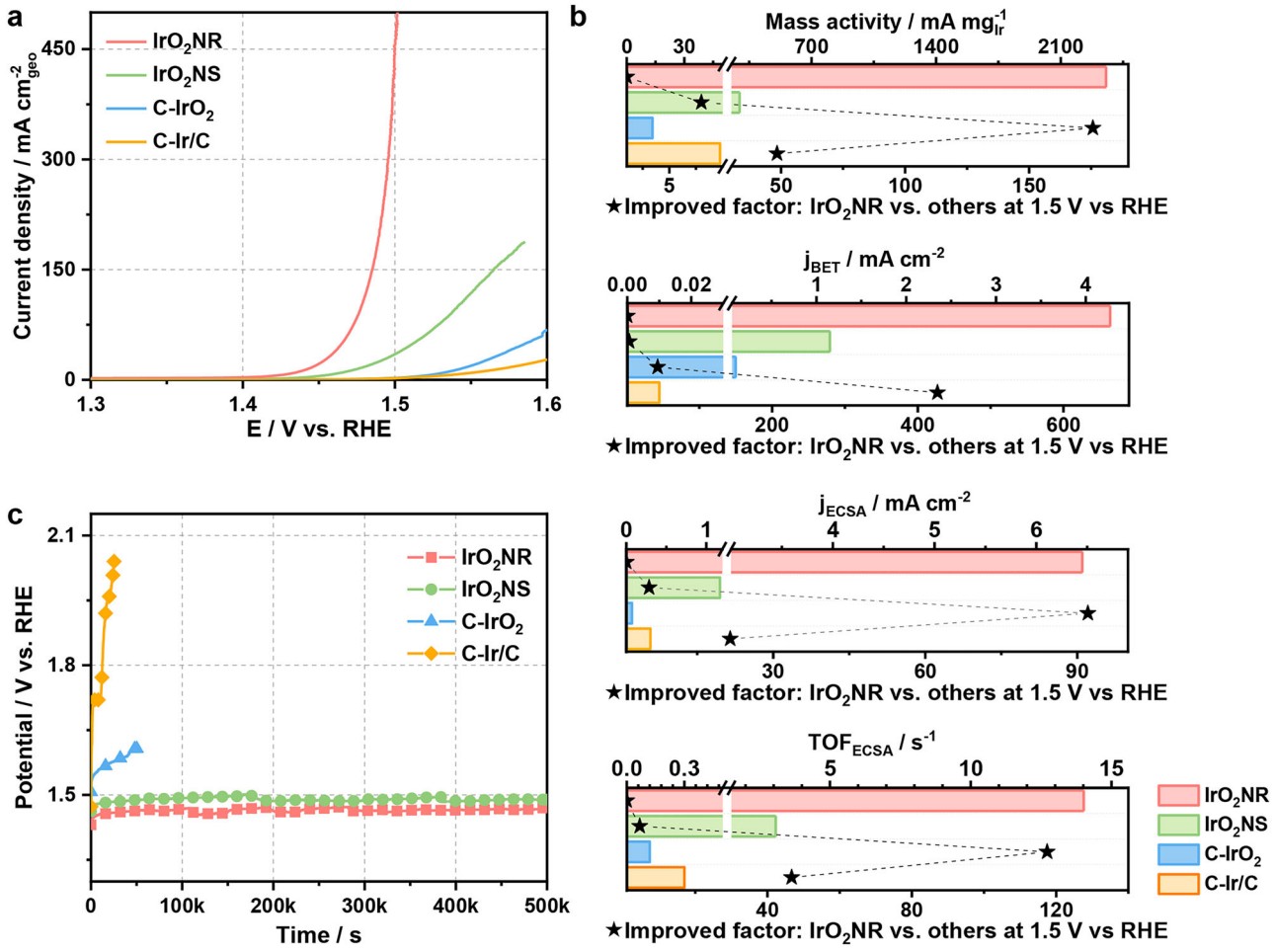

**Fig. 3 | The OER activities of the IrO₂NR, IrO₂NS, C-IrO₂ and C-Ir/C in 0.5 M H₂SO₄.** **a** LSV curves of the IrO₂NR, IrO₂NS, C-IrO₂ and C-Ir/C samples. **b** Mass activities and specific activities based on the BET areas and ECSA areas determined by the mercury underpotential deposition method and TOF values based on the ECSA values at potential of 1.5 V vs. RHE. **c** Chronopotentiometric curves of the IrO₂NR, IrO₂NS, C-IrO₂ and C-Ir/C samples at a constant current density of 10 mA cm⁻².

rutile structure, and that Ir in Ir/C has a face-centred cubic structure. Samples obtained at different steps, including $K_{0.25}IrO_2$ and IrO₂NS, are also employed for comparison. A full connection between the conductive substrate (glassy carbon electrode), IrO₂NR and electrolyte is required to perform electrochemical measurements. To verify the connection phenomena, a contact angle experiment is conducted. IrO₂NR and C-IrO₂ are pressed into tablets. Then, the contact angle of each tablet is tested (Supplementary Fig. 24). Interestingly, the H₂SO₄ (0.5 M) droplet can permeate the IrO₂NR (Supplementary Fig. 24a, c) faster than it can permeate C-IrO₂ (Supplementary Fig. 24b, d), which may be because the IrO₂NR can form a network structure and exhibit good electrical contact between the conductive substrate and the electrolyte. Linear sweep voltammetry (LSV) curves are obtained at a scan rate of 5 mV s⁻¹ with 95% *i*R-compensation (Fig. 3a). The IrO₂NR exhibits excellent OER activity with low overpotential of 205 mV to achieve a current density of 10 mA cm⁻², which is 98 and 117 mV lower than those of C-IrO₂ and C-Ir/C, respectively. $K_{0.25}IrO_2$ is a poor OER catalyst with an overpotential of 344 mV at 10 mA cm⁻² (Supplementary Fig. 25). The OER performance of the IrO₂NS is compared to that of IrO₂NR. IrO₂NS exhibits a high OER activity, with an overpotential of 235 mV@10 mA cm⁻², which is only 30 mV lower than that of IrO₂NR. To comprehensively analyse the catalytic kinetics of OER, the Tafel slopes of the four catalysts are calculated, and they are displayed in Supplementary Fig. 26. The Tafel slope of the IrO₂NR is 46.2 mV dec⁻¹, which is smaller than those of the IrO₂NS (54.2 mV dec⁻¹), C-IrO₂

(61.2 mV dec⁻¹), and C-Ir/C (97.3 mV dec⁻¹), indicating the fast OER kinetics for the IrO₂NR.

A comparison of the overpotential and Tafel slope indicates that IrO₂NRs are more beneficial for OER catalysis. The performance of $K_{0.25}IrO_2$ is too poor for further comparison. To explore the origin of the high activity level, the specific surface areas and electrochemically active surface areas (ECSAs) are analysed by a Brunauer–Emmett–Teller (BET) method and a mercury underpotential deposition method, respectively[45]. The results are shown in Supplementary Figs. 27 and 28, and the data are listed in Supplementary Table 2. The BET area of the IrO₂NR is 47.3 m² g⁻¹, which is larger than those of the other IrO₂ catalysts. The BET area of C-Ir/C is much larger than that of the IrO₂ catalysts due to the addition of carbon black. The ECSAs are calculated to show the catalytic active sites of the catalysts, where the ECSA of the IrO₂NR is 31.3 m² g⁻¹, which is larger than those of C-IrO₂ (15.5 m² g⁻¹) and the IrO₂NS (21.6 m² g⁻¹).

The current densities at 1.5 V vs. RHE based on different standards are calculated and shown in Fig. 3b. The detailed data are listed in Supplementary Table 3. The mass activity, specific activity and turnover frequency (TOF) values of the IrO₂NR are obviously larger than those of C-IrO₂ and C-Ir/C. The mass activity of the IrO₂NR at 1.5 V vs. RHE is 2354.5 mA mg$_{Ir}$⁻¹, which is 175.7 times larger than that of C-IrO₂ (13.4 mA mg$_{Ir}$⁻¹). The specific activities based on the BET and ECSA of the IrO₂NR are 42.7 and 92.1 times larger than those of C-IrO₂. Notably, the performance of the IrO₂NR is better than that of the IrO₂NS.

The mass activity of the IrO$_2$NR is 8.0 times larger than that of the IrO$_2$NS (295.7 mA mg$_{Ir}^{-1}$). The specific activities based on the BET and ECSA of the IrO$_2$NR are also 3.7 and 5.5 times larger than those of the IrO$_2$NS. On the basis of ECSA, a high TOF of 14.01 s$^{-1}$ is achieved by the IrO$_2$NR at 1.5 V vs. RHE, indicating the high quality of its active sites. The OER performance of the IrO$_2$NR is compared to those of other Ir-based catalysts, as shown in Supplementary Table 4. The specific activity levels of the IrO$_2$NR normalized to the BET and ECSA areas are compared (Supplementary Table 5), demonstrating the excellent OER activity in Ir-based catalysts. The Faradic efficiency (FE) is a critical indicator that shows the reaction route. The produced oxygen of IrO$_2$NR at a current density of 20 mA cm$^{-2}$ is collected at 30-min intervals (Supplementary Fig. 29). The FEs of IrO$_2$NR in the 120 min test are maintained above 95.0%, which suggests that the observed catalytic current originates from the water oxidation process.

In addition to activity, stability is an important index for catalysts in real applications, especially in acidic environments. The IrO$_2$NR shows high stability towards the OER (Fig. 3c). The C-IrO$_2$ and C-Ir/C lose their activity after 50000 s at 10 mA cm$^{-2}$, while the activity loss of the IrO$_2$NR is negligible, which is in accordance with the performance of the IrO$_2$NS. The overpotential only increases by ~1.6% for the IrO$_2$NR after 500000 s in 0.5 M H$_2$SO$_4$. The stability of the IrO$_2$NR catalyst is tested at large current densities, including 50 mA cm$^{-2}$ and 100 mA cm$^{-2}$, as shown in Supplementary Fig. 30a. The results indicate the long-term durability of the IrO$_2$NR. A descriptor of the S-number introduced by Geiger et al.[18] is an efficient parameter for evaluating the stability of a catalyst. The S-number is defined as the number of produced oxygen molecules (n(O$_2$)) per number of dissolved iridium ions (n(Ir)) for OER catalysts. IrO$_2$NRs at different current densities (10, 20, 50 and 100 mA cm$^{-2}$) and at different applied potentials (1.40 1.45, 1.50 and 1.55 V vs. RHE) are tested for 3600 s, respectively, and the S-number is calculated as shown in Supplementary Fig. 30b and listed in Supplementary Table 6. The IrO$_2$NR provides more stability than other reported iridium-based OER catalysts (Supplementary Fig. 31)[18,41,46–48]. The morphology, crystal structure and element state of the IrO$_2$NR after the stability test are shown in Supplementary Fig. 32 to evaluate its stability. The layered structure of the IrO$_2$NR is maintained, as confirmed by the XRD pattern (Supplementary Fig. 32a). The SEM and TEM images in Supplementary Fig. 32b, c reveal that the nanoribbon structure of the IrO$_2$NR does not change. The Ir and O elements are uniformly dispersed in the IrO$_2$NR (Supplementary Fig. 32d–f) with the new phase confirmed by the HRTEM image in Supplementary Fig. 32g. XPS spectra of the IrO$_2$NR (Supplementary Fig. 32h, i) indicate that the Ir 4$f$ peak in the IrO$_2$NR maintains the +4 oxidation state. The thickness of the IrO$_2$NR remains in the range of about 6.0 to 12.0 nm, as confirmed by AFM in Supplementary Fig. 33. These data all demonstrate the high stability of the IrO$_2$NR.

### Theoretical analysis of the OER on IrO$_2$NRs

To reveal the underlying origin of the distinguished OER performance, we conduct density functional theory (DFT) calculations to investigate the OER processes on IrO$_2$NR. Here, a theoretical IrO$_2$NR structure model is constructed according to the experimental results, in which the (010) direction is its growth direction. As shown in Fig. 4a, a series of theoretical calculations suggest that the (100) surface contributes to the main OER active sites of the IrO$_2$NR, while that of the conventional Rutile IrO$_2$ is located on the (110) surface. During the DFT calculation process, the active Ir atoms are exposed. For Rutile IrO$_2$ (110), the coordinatively undersaturated Ir site is the reactive site to catalyse water into oxygen, and it does the same to the IrO$_2$NR. The main difference originates from the different geometric configurations of the Ir-O octahedron. An edge-edge sharing mode of IrO$_2$NR, instead of the edge-corner sharing mode in Rutile IrO$_2$ (Fig. 4b, c and Supplementary Fig. 34), is helpful for weakening the adsorption of the OER

intermediates[49]. In addition, a Pourbaix diagram is created to determine the surface termination of the IrO$_2$NR under acidic conditions[50,51]. As shown in Supplementary Fig. 35, at $U = 1.50$ V vs. RHE, the fully oxygen-terminated IrO$_2$ surface is the most stable exposed surface at pH = 0, which is consistent with the experimental conditions. As shown in Fig. 4d, the calculated potential determining step (PDS) of the Rutile IrO$_2$ (110) surface occurs during the O-OH coupling process, namely *O + H$_2$O → *OOH + H$^+$ + e$^-$, with a maximum free energy barrier of 1.89 eV, while the energy change at the PDS of O$_2$ formation (*OOH → * + O$_2$ + H$^+$ + e$^-$) for the IrO$_2$NR decreases to 1.57 eV. Clearly, the obtained theoretical overpotential (η) of 0.34 V for IrO$_2$NR is lower than that of for Rutile IrO$_2$ (110) (0.66 V). The detailed electronic properties of the Ir atoms in the Rutile IrO$_2$ and the IrO$_2$NR are further analysed. As shown in Fig. 4e and Supplementary Figs. 36 and 37, the exposed Ir atoms of IrO$_2$NR have lower $d$ band (or higher O 2$p$ band) centre than those of Rutile IrO$_2$ during the OER cycles. This phenomenon contributes to the weak adsorption of the OER intermediates (Supplementary Fig. 38), self-adjusting the four-electron OER processes to a balanced free energy profile with a low overpotential. As reported, Rutile IrO$_2$ tends to strongly bind O-based intermediates and results in low OER activity due to its high $d$ band centre (or lower O 2$p$ band)[52–54]. In our system, IrO$_2$NR has a low $d$ band centre, which may mainly contribute to its high OER activity.

## Discussion

Layered metal oxides belong to a unique class of functional materials and have great potential in a broad range of applications. Most recent reports focus on two-dimensional (2D) layered materials with nanosheet morphology. However, it should be noted that the nanoribbon layered materials offer a number of advantages similar to or even better than those of 2D nanosheet materials, such as a high surface area, the facilitation of electrical transport, and a natural geometry for in situ probing. Therefore, combining the advantages of the nanoribbon architectures and layered structures may have a significant impact on future applications. Through rational design and fabrication techniques, this combination may lead to an unusual phase that does not exist in bulk materials, thus generating new active sites for electrocatalytic energy conversation reactions. In this study, we obtain a metastable metal oxide with a nanoribbon morphology and discover the essential reason for its OER activity. In this case, IrO$_2$NR is fabricated by a strong alkaline-assisted mechano-thermal process. The structure was determined to be monoclinic phase with a space group of C2/m (12), which evolved from monoclinic K$_{0.25}$IrO$_2$. The IrO$_2$NR exhibits a superior OER activity level with an overpotential of 205 mV at 10 mA cm$^{-2}$ in 0.5 M of H$_2$SO$_4$, which is 98 mV less than that of C-IrO$_2$. Monoclinic IrO$_2$NR shows a very high stability. The high activity level can be maintained for 500000 s during the chronopotentiometry test. In addition to the greater number of exposed active sites at the edges, the Ir atoms in monoclinic IrO$_2$NR have a lower $d$ band energy level than that of rutile phase IrO$_2$, leading to a weaker adsorption of *O in the OER intermediates and to the self-adjusting of the four-electron OER processes to a balanced free energy profile with a low overpotential. With the rapid development of the fields of surface science, nanoscience and nanotechnology, as well as the progress of theoretical research, well-defined metal oxide ribbon nanostructures will pave the way for the design of the next-generation of solid catalysts and provide a deep understanding of structure-activity relationships.

## Methods

### Chemicals

Iridium trichloride (IrCl$_3$, 99.9%) and Nafion solution (5 wt%) were obtained from Alfa Aesar Co. Potassium hydroxide (KOH, 99%) and isopropanol (99.8%) were purchased from Sinopharm Chemical Reagent Co. Commercial iridium oxide (C-IrO$_2$, 99%, rutile phase) was purchased from Aladdin Chemical Reagent Co.

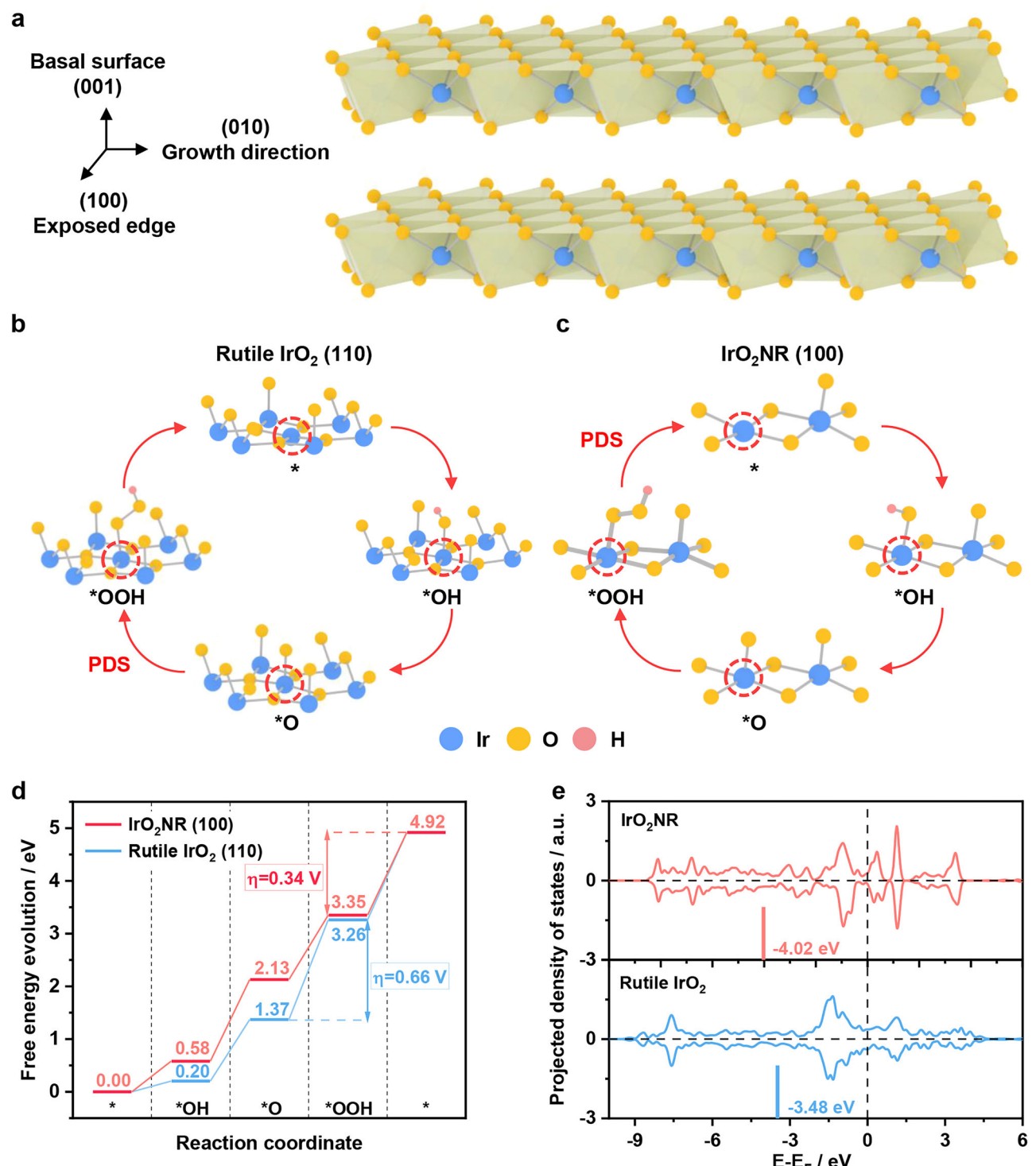

**Fig. 4 | Theoretical analysis of the OER for the IrO$_2$NR. a** Crystal structure of the IrO$_2$NR with an exposed edge for the (001) plane during theoretical calculation. The reaction mechanisms of the OER over the (**b**) Rutile IrO$_2$ (110) and (**c**) IrO$_2$NR (100) samples. **d** Corresponding 4e$^-$ thermodynamic diagram of free energy evolution.

**e** Comparison of the *d*-orbital distribution of the Ir atoms in the Rutile IrO$_2$ and IrO$_2$NR samples. Blue, yellow, and pink balls represent Ir, O and H elements, respectively.

Commercial Ir/C (C-Ir/C with Ir of 20 wt%) was purchased from Premetek Co. The reagents used were of analytical grade.

**Synthesis of layered IrO$_2$NR**

First, an ethanol solution of IrCl$_3$ (3 mg mL$^{-1}$, 100 mL) and KOH aqueous solution (KOH, 9 M, 300 mL) were stirred continuously at 150 °C in a Teflon mortar to form a uniform dark blue slurry. Then, the slurry

was transformed into a homemade mechano-thermal corundum reactor that was fixed in a muffle furnace. The reactor was heated to different temperatures (300–700 °C) by stirring for 2 h. After cooling to room temperature, the samples were washed with double-distilled water and dried by lyophilization. When the heating temperature was 700 °C, the IrO$_2$NR samples were obtained. The fabrication process is schematically shown in Supplementary Fig. 1.

## Material characterization

The morphologies of the samples were observed by scanning electron microscopy (SEM) on a Zeiss Supra 55 with an accelerating voltage of 10 kV. Transmission electron microscopy (TEM), high-resolution TEM (HRTEM), high-angle annular dark-field scanning TEM (HAADF-STEM) and elemental mapping images were further obtained by a TALOS transmission electron microscope with an accelerating voltage of 200 kV. Scanning transmission electron microscopy (STEM) images were collected on a fifth order aberration-corrected transmission electron microscope (JEOL ARM200CF) at 80 kV. X-ray diffraction patterns were collected by an X-ray powder diffractometer (XRD; Philips X'pert PRO MPD) equipped with Cu Kα radiation ($\lambda = 0.15406$ nm). The $IrO_2NR$ sample was laid on the surface of the silicon substrate and measured by powder diffractometer. The silicon substrate has no diffraction peaks. The chemical compositions of the catalysts were measured by X-ray photoelectron spectroscopy (XPS; Kratos AXIS UltraDLD) using Al Kα radiation (1486 eV). The measured binding energies were corrected based on the C 1s energy at 284.6 eV. The surface topographic height of $IrO_2NR$ was measured via atomic force microscopy (AFM; Bruker Dimension Icon). Synchrotron X-ray absorption spectroscopy data were collected at Shanghai Synchrotron Radiation Facility (SSRF, 14 W). Ir L-edge X-ray absorption spectroscopy (XAS) was conducted in transmission mode using the Shanghai Synchrotron Radiation Facility (SSRF, 14 W), China. Pressed pellet specimens were made with mixtures of Ir-based sample powders with boron nitride. The absorption edge jump was optimized from 0–0.5. Energy calibration was conducted by using standard metal (Pt) foil. The reference spectra were used to align the sample spectra to rule out any systematic energy drifts while performing the measurements. The obtained XAS data were analysed using Athena software according to standard procedures using the Demeter program package (Version 0.9.26). The BET specific surface areas were characterized by an American Micromeritics ASAP-2020 porosimeter. The oxygen produced during the OER process was detected by a gas chromatograph (GC Agilent 7890B).

## Electrochemical measurements

All OER experiments were conducted on a CHI 760D electrochemical workstation with a standard three-electrode system. A modified glassy carbon electrode (GCE) and an SCE were chosen for the working electrode and the reference electrode, respectively. GCE has a mirror-like smooth surface with the diameter of 3 mm. A carbon rod (cylindrical carbon material with 3 mm in diameter and 5 cm in length) was selected as the counter electrode.

The catalyst solution was prepared as follows: 4 mg of the catalyst was added to the mixed solution (900 μL of isopropanol and 100 μL of 0.5 wt.% Nafion solution) and ultrasonicated to form a homogenous ink. A 4 μL dispersion was dropped on the surface of the GCE (mass loading: 169.5 mg $cm_{Ir}^{-2}$ for $IrO_2$ and 45.3 mg $cm_{Ir}^{-2}$ for Ir/C) and dried naturally, which was the modified GCE as the working electrode for electrochemical testing. The linear sweep voltammetry (LSV) curves for OER tests were analysed in $O_2$-saturated 0.5 M $H_2SO_4$ with 95% $iR$ correction. The value of compensation resistance (R) was obtained by electrochemical workstation and then the $iR$ correction of the LSV curves was by manual. The stability tests were without $iR$ correction. LSV with a scan rate of 5 mV s⁻¹ was conducted. The stability test was performed by collecting the electrocatalyst dropped on the carbon paper.

The calculation of the S-number was as follows: The $IrO_2NR$ catalyst ink (50 μL, 0.2 mg) was dropped on carbon paper (1 cm × 1 cm) and tested by the chronopotentiometry method at current densities of 10, 20, 50 and 100 mA cm⁻² and applied potentials of 1.40, 1.45, 1.50 and 1.55 V vs. RHE, respectively, for 3600 s. The S-numbers were estimated from the amount of produced oxygen ($n(O_2)$) by assuming 95% FE; the values were divided by the integrated amount of dissolved Ir

ions ($n(Ir)$) under steady-state conditions. The values of $n(Ir)$ were tested by inductively coupled plasma source mass spectrometry (ICP-MS).

The electrochemical surface areas (ECSA) of the Ir-based catalysts were determined by mercury (Hg) underpotential deposition[45]. The corresponding procedures were listed as follows: 0.8 mg catalyst was added to the mixed solution (900 μL of isopropanol and 100 μL of 0.5 wt.% Nafion solution) and ultrasonicated to form a homogenous catalyst ink. The above dispersion (4 μL) was dropped on the GCE (3 mm in diameter) and dried naturally for testing. CV curves were tested in 0.1 M $HClO_4$ solution containing 1 mM mercury nitrate with a potential range from -0.3–0.7 V vs Ag/AgCl. The scan rate was 100 mV s⁻¹. The corresponding calculation equation was as follows:

$$ECSA = \frac{Q}{C} = \frac{S_{peak}}{C\nu} \tag{1}$$

where $C$ is the coulombic charge of 138.6 μC $cm_{Ir}^{-2}$[45]. $S_{peak}$ is the integral area of the adsorbed mercury in the CV curve, and $\nu$ is the scan rate of 100 mV s⁻¹.

The turnover frequencies (TOFs) of the electrocatalysts were defined as the produced oxygen by the moles of the active sites per unit time. The TOFs were calculated as follows[55]:

$$TOF_{BET/ECSA} = \frac{(1.56 \times 10^{15} \frac{O_2/s}{cm^2} \text{ per } \frac{mA}{cm^2}) \times j}{(\text{active sites}) \times A_{BET/ECSA}} \tag{2}$$

The average active surface atoms per square centimetre of the $IrO_2NRs$ were calculated as follows:

$$\text{Active sites}_{(IrO_2NR)} = \frac{1\,\text{atom}}{(4.43 \times 3.14)\text{Å}^2} = 7.18 \times 10^{14} \frac{\text{atoms}}{cm^2} \tag{3}$$

The average active surface atoms per square centimetre of C-$IrO_2$ were calculated as follows:

$$\text{Active sites}_{(C-IrO_2)} = \frac{2\,\text{atom}}{(4.498 \times 3.154)\text{Å}^2 \times \sqrt{2}} = 9.97 \times 10^{14} \frac{\text{atoms}}{cm^2} \tag{4}$$

The average active surface atoms per square centimetre of C-Ir/C were calculated as follows:

$$\text{Active sites}_{(C-Ir/C)} = \frac{0.5\,\text{atom}}{0.5 \times (\sqrt{2}/2 \times 3.839 \times \sqrt{2}/2 \times 3.839)\text{Å}^2 \times \sin 60°}$$
$$= 15.67 \times 10^{14} \frac{\text{atoms}}{cm^2} \tag{5}$$

The Faradaic efficiency (FE) was obtained according to the following equation:

$$FE = 4nF/It \times 100\% \tag{6}$$

where $F$ is the Faraday constant (96485 C mol⁻¹), $n$ is the number of moles of the produced oxygen, $I$ is the current (A) and $t$ is the reaction time (s).

## DFT calculations

All theoretical simulations were implemented under the framework of density functional theory for the geometric and electronic analyses; their concrete realization relied on the Vienna ab-initio Simulation Package in version 5.4.1[56,57]. The description of electronic exchange-correlation energy adopted the revised Perdew-Burke-Ernzerhof function due to its more accurate treatment of the surface adsorption system[58]. Constant charge model (CCM) was

used during DFT calculation in neutral cells. The electronic cut-off energy was set to 520 eV and the convergence thresholds of energy and force during geometry optimization corresponded to $10^{-4}$ eV and $-0.03$ eV $Å^{-1}$, respectively. For $IrO_2NR$, we used the Gamma-centred Monkhorst-Pack $4 \times 1 \times 1$ k-point mesh[59], where 4-kpoints were sampled along the growth direction. As for the Rutile $IrO_2$ slab model, 4-unit-thick $2 \times 1$ (110) surface was cleaved from its bulk counterpart for calculation and a $3 \times 4 \times 1$ k-point mesh was sampled. An implicit solvation model[60] on top of a monolayer water covered surface was constructed (Supplementary Fig. 39) for the calculation of Pourbaix diagram. We analyse the Pourbaix diagram to determine the equilibrium surface under reaction conditions, and a four-layered (100) surface was cleaved from its bulk counterpart with full oxygen-termination for further oxygen evolution processes. The free energy evaluation of the four-electron OER was based on a computational hydrogen electrode model[61,62], in which the free energy of the electron-proton pair was approximated to half that of hydrogen at room temperature. The OER pathway follows a widely accepted adsorbate evolution mechanism as shown in Eqs. (7)-(10).

$$H_2O + * \rightarrow OH* + (H^+ + e^-) \tag{7}$$

$$OH* \rightarrow O* + (H^+ + e^-) \tag{8}$$

$$H_2O + O* \rightarrow OOH* + (H^+ + e^-) \tag{9}$$

$$OOH* \rightarrow O_2 + * + (H^+ + e^-) \tag{10}$$

The change of the Gibbs free energy (ΔG) for each intermediate is defined by $\Delta G = \Delta E + \Delta E_{ZPE} - T\Delta S$, where $\Delta E_{ZPE}$ is the difference in zero-point energy between the adsorbed and the gas phase, and T is the temperature of 300 K and ΔS is the entropy change. The theoretical overpotential was determined by $\eta = \max\{G_i\} - 1.23$, where $G_i$ represents the free energy change of every OER step. The simulated coordinates for $IrO_2NR$ and Rutile $IrO_2$ were supplied in Supplementary Notes 1 and 2 and the code input file was in Supplementary Note 3. The zero-point vibrational energies (ZPVEs) and entropic contributions were given in Supplementary Table 7.

## Data availability

The data generated in this study are provided in the Supplementary Information/Source Data file. Source data are provided with this paper.

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

## Acknowledgements

This work was financially supported by the National MCF Energy R&D Program (2018YFE0306105), National Key R&D Program of China (2020YFA0406104, 2020YFA0406101), Innovative Research Group Project of the National Natural Science Foundation of China (51821002), National Natural Science Foundation of China (51902217, 21905188, 51725204, 21771132, 51972216, 52041202), Innovative Research Group Project of the National Natural Science Foundation of China (51821002), the major project of Basic Science (natural science) of Jiangsu Province (21KJA430001), Jiangsu Provincial Natural Science Foundation (BK20211316), the Suzhou Municipal Science and Technology Bureau (SYG202125), State Key Laboratory of Physical Chemistry of Solid Surfaces, Xiamen University (202113), Natural Science Foundation of Jiangsu Province (BK20190041), China Postdoctoral Science Foundation Grant (2019M651937), the project of scientific and technologic infrastructure of Suzhou (SZS201708), the Priority Academic Program Development of Jiangsu Higher Education Institutions (PAPD). Collaborative Innovation Center of Suzhou Nano Science & Technology, and the 111 Project. We also acknowledge the support from Shanghai Synchrotron Radiation Facility for the XAS experiments.

## Author contributions

Q.S. supervised the research. F.L., W.Z., Z.F., M.S., Z.K. and Q.S. performed most of the experiments and data analysis. J.Z. performed the XAS experiment. K.Y., Y.J. and Y.L. performed and analysed the DFT simulations. F.L., Q.S., M.S. and Z.K. wrote the paper. All authors discussed the results and commented on the manuscript. F.L., K.Y. and Y.J. equally contribute to this work.

## Competing interests

The authors declare no competing interests.
