## [Peer Review File · Nature Communications]

Iridium Oxide Nanoribbons with Metastable Monoclinic Phase for Highly Efficient Electrocatalytic Oxygen EvolutionEditorial Note: This manuscript has been previously reviewed at another journal that is not operating a transparent peer review scheme. This document only contains reviewer comments and rebuttal letters for versions considered at Nature Communications.

Reviewers' Comments:

Reviewer #3:

Remarks to the Author:

Overall, I feel that the reviewer comments were well addressed with significant additions since last review (I reviewed the paper 2 times for Nature Catalysis). I recommend the manuscript for publication in Nature Communications after minor revision:

1. Reviewer 3, question 2: there is no statement in the text emphasizing that the glassy carbon was smooth and porosity does not play a role. Please add this to the manuscript.
2. Additional questions (please clarify in the manuscript). What is a carbon rob? (line 377)? How was the glassy carbon electrode modified (line 375-376)?
3. Rev3, Q4: The technique "HRTEM" refers to atomic resolution at high magnification, which is not seen in Supp Fig 12. Please either present a real HRTEM image, or change the technique name to "TEM" in the modified content.
4. Rev3, Q5: why is the (402) ring in SAED so strong whereas in the XRD reflection at 40.2 deg one of the weakest? Please clarify in the manuscript.
5. Rev3, Q6 and Q8: please state these replies in the manuscript. These clarifications are very important - not only for the reviewers, but also for the readers!
6. Rev3, Q7: the normal-normal plot is fine, but the Gaussian function should be removed, as the distribution is asymmetric.
7. Figure 4 (c): The IrO₂NR(100) exposed edge does not look correctly depicted: from Figure 4 (a) I see 3 oxygens per Ir terminating the surface. The caption for Fig. 4 (b and c) is a reaction mechanism, not a free energy profile.
8. ESI references: please use full author lists.

Reviewer #4:

Remarks to the Author:

The manuscript "Iridium Oxide Nanoribbons with Metastable Monoclinic Phase for Highly Efficient Electrocatalytic Oxygen Evolution" by Fan Liao et al. provides a new morphology with the enhanced catalytic performance. The search and design of new phases is crucial for the future development of efficient electrocatalysts. However, there are some major comments related to the DFT calculations that should be addressed, i.e. the description of methods in computational methodology part is not sufficient to reproduce the obtained results.

1. There is no information on the rutile slab at all. What are the cell dimensions, what is the k-point mesh used in this study? Providing the coordinates for both rutile and IrO₂NR will be helpful for readers' understanding.

2. Currently the number of OER mechanisms are being discussed in literature, such as AEM, LOM, I2M. According to Fig. 4, the authors considered well-established AEM mechanism using computational hydrogen electrode (CHE). However, on page 13, line 292 they talk about O-O coupling: "the calculated potential determining step (PDS) of the Rutile IrO₂ (110) surface occurs during the O-O coupling process with a maximum free energy barrier of 1.89 eV" and refer to Figure 4d. This sounds confusing, O-O coupling is related to I2M mechanism and I assume it was not considered in this study. The presented diagram in Figure 4d shows the four electron-proton coupled steps in AEM. This should be clarified. Additionally, all equations on the calculated Gibbs free energies of reaction intermediates should be provided in methodology part, including ZPVE and entropic contributions.
3. The equation for calculating the theoretical OER overpotential should be corrected. OER overpotential is defined by the largest energy step – 1.23 V.
4. The authors provide the computed Pourbaix diagrams, showing fully oxidized surface as the most favorable under high potential. They also discussed that O-terminated surface was used to calculate the theoretical overpotentials, which is reasonable. However, they include the implicit solvation considering water adsorbed on the surface "An implicit solvation model⁶¹ on top of a monolayer water covered surface was adopted to consider four-electron OER processes (Supplementary Fig. 36)." Therefore, it is not clear if they used O-terminated surface (Fig. 4d) or H₂O-covered surface (Suppl. Fig. 36) to calculate the OER overpotentials. Are the presented overpotentials calculated in gas phase or implicit solvent?
5. The authors used constant-potential simulations in JDFTx, setting up constant potential to 1.23 V. What surface coverage is used? According to Pourbaix diagram, the surface should be OH-terminated at these conditions.
6. The authors rely on d band center approach, however the O 2p band centers and covalency are more established OER descriptors [Chem. Mater. 2019, 31, 785–797; <https://doi.org/10.1038/s41929-020-0465-6>] and worth checking.
7. Also, it is known that OER overpotentials are sensitive to the chosen functionals and codes [<https://doi.org/10.1002/cctc.201601662>]. Therefore, I suggest sticking to one chosen approach and not to compare the results from different software.

Additionally, some typos through the text can be corrected, such as "center", "analyze", as well as naming at Supplementary Figure 32.

Overall, the computational methodology part should be carefully rewritten. I recommend including the code input files used in this study in Supplementary Materials, as well as atomic coordinates.

Response to the Referees' Comments

Dear Reviewers,

Thank you for your precious time and the constructive comments on our manuscript titled "Iridium Oxide Nanoribbons with Metastable Monoclinic Phase for Highly Efficient Electrocatalytic Oxygen Evolution" (NCOMMS-22-39053). We sincerely appreciate your comments and suggestions on our work, which definitely improve our manuscript. According to all the comments, we have made a detailed response and substantial revisions in our revised manuscript.

Your comments are shown in black and our responses are shown in blue. The changes in the manuscript are indicated in red.

Reviewer #3 (Remarks to the Author):

Overall, I feel that the reviewer comments were well addressed with significant additions since last review (I reviewed the paper 2 times for Nature Catalysis). I recommend the manuscript for publication in Nature Communications after minor revision.

[Author's Response]: We would like to thank you for your precious time and your confirmation about our previous version. Your constructive suggestions lead to further improvement of the quality for our work.

1. Reviewer 3, question 2: there is no statement in the text emphasizing that the glassy carbon was smooth and porosity does not play a role. Please add this to the manuscript.

[Author's Response]: Thank you for your valuable suggestion. We added the description of the glassy carbon electrode in the part of "Electrochemical Measurements".

[Modified content]: [Line 382, Page 17] "GCE has a mirror-like smooth surface with the diameter of 3 mm."

2. Additional questions (please clarify in the manuscript). What is a carbon rod? (line 377)? How was the glassy carbon electrode modified (line 375-376)?

[Author's Response]: Thank you. First, the counter electrode used is a carbon rod, which has the cylindrical shape with the diameter of 3 mm and the length of 5 cm.

In addition, the catalyst ink dropped on the surface of the glassy carbon electrode is defined as the modified glassy carbon electrode. In details, 4 mg of the catalyst was added to the mixed solution (900 μ L of isopropanol and 100 μ L of 0.5 wt.% Nafion solution) and ultrasonicated to form a homogenous ink. A 4 μ L dispersion was dropped on the surface of the glassy carbon electrode and dried naturally, which is the modified glassy carbon electrode for electrochemical testing.

[Modified content]: [Line 382, Page 17] "A carbon rod (cylindrical carbon material with 3 mm in diameter and 5 cm in length) was selected as the counter electrode."

[Line 387, Page 17] "A 4 μ L dispersion was dropped on the surface of the GCE and dried naturally, which was the modified GCE as the working electrode for electrochemical testing."

3. Rev3, Q4: The technique "HRTEM" refers to atomic resolution at high magnification, which is not seen in Supp Fig 12. Please either present a real HRTEM image, or change the technique name to "TEM" in the modified content.

[Author's Response]: Thank you for your valuable suggestion. The real HRTEM image of the initial slurry is difficult to be obtained due to the existing strong alkali. Following your suggestion, we have changed the technique name to "TEM".

[Modified content]: [Line 135, Page 7] “The slurry sample shows a droplet-like morphology as seen in the TEM image in **Supplementary Fig. 14b**.”

4. Rev3, Q5: why is the (402) ring in SAED so strong whereas in the XRD reflection at 40.2 deg one of the weakest? Please clarify in the manuscript.

[Author’s Response]: Thank you for your valuable suggestion. We retested the HRTEM and SAED of $K_{0.25}IrO_2$. The deduced (402) ring in SAED in previous version is unreliable because the scale bar of SAED from the TEM instrument was not corrected. After it has been calibrated, the correct SEAD pattern was obtained. As seen in the HRTEM image in **Supplementary Fig. 16a**, the crystal plane of (211) of $K_{0.25}IrO_2$ is marked in the HRTEM image. At the same time, the (211) plane can be detected in the SAED pattern (**Supplementary Fig. 16b**), which is consistent with the HRTEM image. The image with the origin scale bar given by instrument is also provided in **Supplementary Fig. 16b**. We have also changed the supplementary figure and related description.

[Modified content]: [Line 144, Page 7] “The HRTEM image (**Supplementary Fig. 16a**) shows that the crystal plane of (211) of $K_{0.25}IrO_2$. The SAED pattern shows that $K_{0.25}IrO_2$ is polycrystalline (**Supplementary Fig. 16b**). The bright ring corresponds to the (211) plane (JCPDS No. 85-2185).”

Supplementary Figure 16. Chemical composition analysis of $K_{0.25}IrO_2$ obtained at reaction temperature of 500 °C. (a) HRTEM image, (b) SAED, (c) HAADF-STEM image and corresponding elemental EDS mapping image showing the distributions of (d) Ir, (e) O, and (f) K elements.

5. Rev3, Q6 and Q8: please state these replies in the manuscript. These clarifications are very important - not only for the reviewers, but also for the readers!

[Author’s Response]: Following your suggestion, we have added the replies for Q6 (The XRD

measurement method and the explanation of the preferred orientation) and Q8 (The lattice strain in the IrO₂NR) in the main paper.

[Modified content]: [Line 363, Page 16] “The IrO₂NR sample was laid on the surface of the silicon substrate and measured by powder diffractometer. The silicon substrate has no diffraction peaks.”

[Line 103, Page 5] “Additionally, due to the preferred orientation of the IrO₂NR, no peaks other than the layered structure peaks are observed in the XRD pattern.”

[Line 74, Page 4] “The lattice tension/compression can be observed at the edges of the IrO₂NRs according to the HRTEM images (Supplementary Fig. 5). The average lattice strain is calculated by the Williamson-Hall equation based on the XRD data (Supplementary Fig. 6), which is about 0.388%. Although the lattice strain exists in IrO₂NR, the effect on activity may be limited due to its low value when we compared with the stain in previous reference about IrO₂ for OER catalysis³².”

Supplementary Figure 5. (a) Low magnification and (b) high magnification HRTEM images of IrO₂NR showing the lattice tension/compression.

Supplementary Figure 6. Plot for Williamson-Hall analysis to calculate the lattice strain of IrO₂NR. According to the Williamson-Hall equation: $\beta \times \cos\theta = K\lambda / D + 4\epsilon \times \sin\theta$, where β is the full width at half-maximum of the peak, θ is the Bragg angle, K is the shape factor, D is the crystallite size, and λ is the wavelength of X-ray, the lattice strain of IrO₂NR is calculated to be 0.388%. Error bars have been added in the Williamson-Hall plot based on three XRD test results.

32. Meng, G. et al. Strain regulation to optimize the acidic water oxidation performance of atomic-layer IrO_x. *Adv. Mater.* **31**, 1903616 (2019).

6. Rev3, Q7: the normal-normal plot is fine, but the Gaussian function should be removed, as the distribution is asymmetric.

[Author’s Response]: Thank you for your valuable suggestion. We removed the Gaussian function

fitted line in **Supplementary Fig. 3**.

[Modified content]:

Supplementary Figure 3. Statistics histogram showing the width of IrO₂NR. The width values are obtained by 500 nanoribbons.

7. Figure 4 (c): The IrO₂NR(100) exposed edge does not look correctly depicted: from Figure 4 (a) I see 3 oxygens per Ir terminating the surface. The caption for Fig. 4 (b and c) is a reaction mechanism, not a free energy profile.

[Author's Response]: Thank you for your valuable suggestion. We added an explanation figure in **Supplementary Fig. 34**. For DFT calculation, the exposed Ir atoms (red circle) are regarded as the active sites. When rotating the model in **Supplementary Fig. 34a** along the a-c plane, it shows the model in **Supplementary Fig. 34b**, which is same to the model in **Supplementary Fig. 34c** and **Fig. 4c**.

In addition, the caption for Fig. 4 (b and c) has been modified according to your advice.

[Modified content]:

Supplementary Figure 34. (a) and (b) Crystal structures of the IrO₂NR with an exposed edge for the (001) plane during theoretical calculation with different viewing angles. (c) The ball-and-stick model extracted from (b) to show the surface location of active Ir in Figure 4c.

[Line 643, Page 32] “Figure 4. Theoretical analysis of the OER for the IrO₂NR. The reaction mechanisms of the OER over the (b) Rutile IrO₂ (110) and (c) IrO₂NR (100) samples.”

8. ESI references: please use full author lists.

[Author’s Response]: Thank you. Following your suggestion, full author lists have been added in the references in Supplementary information.

Reviewer #4 (Remarks to the Author):

The manuscript “Iridium Oxide Nanoribbons with Metastable Monoclinic Phase for Highly Efficient Electrocatalytic Oxygen Evolution” by Fan Liao et al. provides a new morphology with the enhanced catalytic performance. The search and design of new phases is crucial for the future development of efficient electrocatalysts. However, there are some major comments related to the DFT calculations that should be addressed, i.e. the description of methods in computational methodology part is not sufficient to reproduce the obtained results.

[Author’s Response]: Thank you for your precious time to put forward constructive suggestions on our manuscript, which are important for the further improvements of our manuscript.

1. There is no information on the rutile slab at all. What are the cell dimensions, what is the k-point mesh used in this study? Providing the coordinates for both rutile and IrO₂NR will be helpful for readers’ understanding.

[Author’s Response]: Thank you for your valuable suggestion. As for the rutile IrO₂ slab model, 4-unit-thick 2×1 (110) surface was cleaved from its bulk counterpart for calculation and a 3×4×1 k-point mesh was sampled.

In addition, following your suggestions, the concrete coordinates of Rutile IrO₂ and IrO₂NR for every OER intermediates (*O, *OH and *OOH) were added in **Supplementary Note 1** and **Supplementary Note 2**.

[Modified content]: [Line 431, Page 19] “As for the Rutile IrO₂ slab model, 4-unit-thick 2×1 (110) surface was cleaved from its bulk counterpart for calculation and a 3×4×1 k-point mesh was sampled.”

2. Currently the number of OER mechanisms are being discussed in literature, such as AEM, LOM, I2M. According to Fig. 4, the authors considered well-established AEM mechanism using computational hydrogen electrode (CHE). However, on page 13, line 292 they talk about O-O coupling: “the calculated potential determining step (PDS) of the Rutile IrO₂ (110) surface occurs during the O-O coupling process with a maximum free energy barrier of 1.89 eV” and refer to Figure 4d. This sounds confusing, O-O coupling is related to I2M mechanism and I assume it was not considered in this study. The presented diagram in Figure 4d shows the four electron-proton coupled steps in AEM. This should be clarified. Additionally, all equations on the calculated Gibbs free energies of reaction intermediates should be provided in methodology part, including ZPVE and entropic contributions.

[Author’s Response]: Thank you for your valuable suggestion. In this work, we only consider the AEM mechanism. The O-O coupling refers to the O-OH coupling at the third proton-electron step, namely *O+H₂O→*OOH+H⁺+e⁻. We have clarified this description in the manuscript. Meanwhile, the ZPVE and entropic contributions of all intermediates were given in **Supplementary Table 7**.

[Modified content]: [Line 296, Page 13] “As shown in **Fig. 4d**, the calculated potential determining step (PDS) of the Rutile IrO₂ (110) surface occurs during the O-OH coupling process, namely *O + H₂O → *OOH + H⁺ + e⁻, with a maximum free energy barrier of 1.89 eV, while the energy change at the PDS of O₂ formation (*OOH → * + O₂ + H⁺ + e⁻) for the IrO₂NR decreases to 1.57 eV.”

[Line 439, Page 20] “The OER pathway follows a widely accepted adsorbate evolution mechanism as shown in equations (7)-(10).

The change of the Gibbs free energy (ΔG) for each intermediate is defined by $\Delta G = \Delta E + \Delta E_{\text{ZPE}} - T\Delta S$, where ΔE_{ZPE} is the difference in zero-point energy between the adsorbed and the gas phase, and T is the temperature of 300 K and ΔS is the entropy change.”

[Line 449, Page 20] “The zero-point vibrational energies (ZPVEs) and entropic contributions were given in **Supplementary Table 7**.”

Supplementary Table 7. ZPVEs and entropic correction at 300 K.

	H₂O	H₂	*O	*OH	*OOH
ZPVE (eV)	0.60	0.30	0.07	0.37	0.46
ΔS (eV)	0.59	0.41	0.09	0.12	0.16

3. The equation for calculating the theoretical OER overpotential should be corrected. OER overpotential is defined by the largest energy step – 1.23 V.

[Author’s Response]: Thank you for your valuable suggestion. Accordingly, we corrected the equation to $\eta = \max\{G_i\} - 1.23$.

[Modified content]: [Line 447, Page 20] “The theoretical overpotential was determined by $\eta = \max\{G_i\} - 1.23$, where G_i represents the free energy change of every OER step.”

4. The authors provide the computed Pourbaix diagrams, showing fully oxidized surface as the most favorable under high potential. They also discussed that O-terminated surface was used to calculate the theoretical overpotentials, which is reasonable. However, they include the implicit solvation considering water adsorbed on the surface “An implicit solvation model⁶¹ on top of a monolayer water covered surface was adopted to consider four-electron OER processes (Supplementary Fig. 36).” Therefore, it is not clear if they used O-terminated surface (Fig. 4d) or H₂O-covered surface (Suppl. Fig. 36) to calculate the OER overpotentials. Are the presented overpotentials calculated in gas phase or implicit solvent?

[Author’s Response]: Thank you for your valuable suggestion. **Supplementary Fig. 36 (Supplementary Fig. 39** in the revised Supplementary information) is the model used to calculate the Pourbaix diagram, from which we found that O-terminated surface is energetic-favorable to calculate the theoretical OER overpotentials as shown in **Fig. 4d**. In addition, the presented overpotentials were calculated in implicit solvent. We revised this discussion to emphasize the function of **Supplementary Fig. 36** and **Fig. 4d** in details.

[Modified content]: [Line 433, Page 19] “An implicit solvation model⁶¹ on top of a monolayer water covered surface was constructed (**Supplementary Fig. 39**) for the calculation of Pourbaix diagram. We analyse the Pourbaix diagram to determine the equilibrium surface under reaction conditions, and a four-layered (100) surface was cleaved from its bulk counterpart with full oxygen-termination for further oxygen evolution processes.”

5. The authors used constant-potential simulations in JDFTx, setting up constant potential to 1.23 V. What surface coverage is used? According to Pourbaix diagram, the surface should be OH-terminated at these conditions.

[Author's Response]: Thank you for your valuable suggestion. In the DFT calculation with constant charge model (CCM), we adopted the model with O-terminated surface. In order to make a better comparison, the model with same surface termination was used for constant-potential simulation in JDFTx.

Based on your Comment 7, we compared the OER processes on IrO₂NR and Rutile IrO₂ based on the RPBE functional and deleted the discussion part with using constant-potential simulations.

6. The authors rely on d band center approach, however the O 2p band centers and covalency are more established OER descriptors [Chem. Mater. 2019, 31, 785–797; <https://doi.org/10.1038/s41929-020-0465-6>] and worth checking.

[Author's Response]: Following your suggestions, we have checked the O 2p band centers and the covalency in IrO₂NR and Rutile IrO₂. As shown in **Supplementary Fig. 37**, the O in IrO₂NR has a higher band center, which leads to a higher OER activity. We have cited this reference [Chem. Mater. 31, 785-797 (2019)] to support our conclusion.

However, the conclusion obtained by the covalency model in our work (**Figure R1**) is adverse from this reference. This may be due to the descriptor is more specific in the spinel oxide structure.

Supplementary Figure 37. Comparison of the 2p-orbital distribution of the O atoms in the Rutile IrO₂ and IrO₂NR.

Figure R1. The covalency of O and Ir in bulk IrO₂NR and Rutile IrO₂.

[Modified content]: [Line 302, Page 13] “As shown in Fig. 4e and Supplementary Figs. 36 and 37, the exposed Ir atoms of IrO₂NR have lower *d*-band (or higher O 2*p* band) centre than those of Rutile IrO₂ during the OER cycles.”

[Line 306, Page 14] “As reported, Rutile IrO₂ tends to strongly bind O-based intermediates and results in low OER activity due to its high *d*-band centre (or lower O 2*p* band)⁵³⁻⁵⁵.”

55. Jacobs, R. Hwang, J. Shao-Horn, Y. & Morgan, D. Assessing correlations of perovskite catalytic performance with electronic structure descriptors. *Chem. Mater.* **31**, 785-797 (2019).

Supplementary Figure 37. Comparison of the 2*p*-orbital distribution of the O atoms in the Rutile IrO₂ and IrO₂NR.

7. Also, it is known that OER overpotentials are sensitive to the chosen functionals and codes [<https://doi.org/10.1002/cctc.201601662>]. Therefore, I suggest sticking to one chosen approach and not to compare the results from different software.

[Author’s Response]: According to the reviewer’s suggestion, we chose function of revised Perdew–Burke–Ernzerhof (RPBE) and constant charge model (CCM) during DFT calculation in this work.

In addition, we deleted the following paragraph that using constant potential model (CPM) from the text: “A constant potential model is also adopted to calculate the free energy barriers of the potential determining steps at 1.23 V vs. RHE⁵³, which provide theoretical overpotentials for IrO₂NR and Rutile IrO₂ of 0.27 and 0.75 V, respectively. Thus, the electrochemical electrode potential shows a similar theoretical trend as the constant charge model.”.

[Modified content]: [Line 425, Page 19] “The description of electronic exchange–correlation energy adopted the revised Perdew–Burke–Ernzerhof function due to its more accurate treatment of the surface adsorption system. Constant charge model (CCM) was used during DFT calculation.”

Additionally, some typos through the text can be corrected, such as “center”, “analyze”, as well as naming at Supplementary Figure 32.

[Author’s Response]: Thank you for your valuable suggestion. The typos are modified according to your advice. The original **Supplementary Fig. 32** has been deleted according to the suggestion of Reviewer #3.

Overall, the computational methodology part should be carefully rewritten. I recommend including

the code input files used in this study in Supplementary Materials, as well as atomic coordinates.

[Author's Response]: Thank you for your valuable suggestion. The computational methodology part was carefully rewritten. The concrete code input files and the atomic coordinates of the OER intermediates on Rutile IrO₂ and IrO₂NR are given in **Supplementary Notes 1-3**. The related files are also uploaded to the submission system.

Reviewers' Comments:

Reviewer #4:

Remarks to the Author:

Overall, the authors have significantly improved the computational part of the paper, and it can be published after minor revisions. I would recommend to remove " To consider the effects of electrochemical electrode potentials, we adopted the software JDFTx63 to calculate the free energy barriers at the PDS under a constant potential model" from the main paper. Additionally, an explanation of chosen ISMEAR = 0 is needed, since IrO₂ is metallic oxide and ISMEAR = 1 is the more appropriate choice. Also, I would suggest to make it clear that the authors used neutral cell in constant charge calculations.

Response to the Reviewer

The reviewer's comments are shown in black and our responses are shown in blue. The changes in the manuscript are indicated in red.

Reviewer #4 (Remarks to the Author):

Overall, the authors have significantly improved the computational part of the paper, and it can be published after minor revisions. I would recommend to remove " To consider the effects of electrochemical electrode potentials, we adopted the software JDFTx63 to calculate the free energy barriers at the PDS under a constant potential model" from the main paper. Additionally, an explanation of chosen ISMEAR = 0 is needed, since IrO₂ is metallic oxide and ISMEAR = 1 is the more appropriate choice. Also, I would suggest to make it clear that the authors used neutral cell in constant charge calculations.

[Author's Response]: We would like to thank you for the positive comments and recommendation of the publication in the Nature Communications. Your comments lead to further improve the quality of our work.

Based on your suggestions, firstly we have deleted the paragraph "To consider the effects of electrochemical electrode potentials, we adopted the software JDFTx63 to calculate the free energy barriers at the PDS under a constant potential model" from the main paper.

Secondly, the reason we chose ISMEAR=0 is mainly based on the VASP official manual: "If you have no a priori knowledge of your system, for instance, if you do not know whether your system is an insulator, semiconductor or metal then always use Gaussian smearing ISMEAR=0 in combination with a small SIGMA=0.03-0.05" [<https://www.vasp.at/wiki/index.php/ISMEAR>]. In order to dispel your concerns, we also compared our results by using ISMEAR=0 and 1 and found that there is no clear difference on final energies and optimized configurations for OER.

Thirdly, we made it clear that a neutral cell is adopted in constant charge calculations.

[Added content]: [Line 420, Page 19] "Constant charge model (CCM) was used during DFT calculation in neutral cells."